# Quantifying ecosystem states and state transitions of the Upper Mississippi River System using topological data analysis

**Danelle Marie Larson**[1]*, **Wako Bungula**[2], **Casey McKean**[3], **Alaina Stockdill**[3], **Amber Lee**[3], **Frederick Forrest Miller**[3], **Killian Davis**[3]

**1** U.S. Geological Survey, Upper Midwest Environmental Sciences Center, La Crosse, Wisconsin, United States of America, **2** University of Wisconsin La-Crosse, Department of Mathematics and Statistics, La Crosse, Wisconsin, United States of America, **3** University of Wisconsin La-Crosse, Research Experience for Undergraduates Program, La Crosse, Wisconsin, United States of America

* dmlarson@usgs.gov

**Data Availability Statement:** All data were publicly accessed on 1 June, 2021 through the Upper Mississippi River Restoration Program's Long-

## Abstract

Aquatic systems worldwide can exist in multiple ecosystem states (i.e., a recurring collection of biological and chemical attributes), and effectively characterizing multidimensionality will aid protection of desirable states and guide rehabilitation. The Upper Mississippi River System is composed of a large floodplain river system spanning 2200 km and multiple federal, state, tribal and local governmental units. Multiple ecosystem states may occur within the system, and characterization of the variables that define these ecosystem states could guide river rehabilitation. We coupled a long-term (30-year) highly dimensional water quality monitoring dataset with multiple topological data analysis (TDA) techniques to classify ecosystem states, identify state variables, and detect state transitions over 30 years in the river to guide conservation. Across the entire system, TDA identified five ecosystem states. State 1 was characterized by exceptionally clear, clean, and cold-water conditions typical of winter (i.e., a clear-water state); State 2 had the greatest range of environmental conditions and contained most the data (i.e., a status-quo state); and States 3, 4, and 5 had extremely high concentrations of suspended solids (i.e., turbid states, with State 5 as the most turbid). The TDA mapped clear patterns of the ecosystem states across several riverine navigation reaches and seasons that furthered ecological understanding. State variables were identified as suspended solids, chlorophyll *a*, and total phosphorus, which are also state variables of shallow lakes worldwide. The TDA change detection function showed short-term state transitions based on seasonality and episodic events, and provided evidence of gradual, long-term changes due to water quality improvements over three decades. These results can inform decision making and guide actions for regulatory and restoration agencies by assessing the status and trends of this important river and provide quantitative targets for state variables. The TDA change detection function may serve as a new tool for predicting the vulnerability to undesirable state transitions in this system and other ecosystems with sufficient data. Coupling ecosystem state concepts and TDA tools can be transferred to any ecosystem with large data to help classify states and understand their vulnerability to state transitions.

term Resource Monitoring (https://umesc.usgs.gov/ltrm-home.html). Our final datasets and analysis scripts are publicly and permanently archived in ScienceBase (DOI: 10.5066/P97O6O6PP).

**Funding:** The salaries of authors Danelle Marie Larson and Wako Bungula, the data collection, data analysis, and decision to publish were funded by the U.S. Army Corps of Engineers' Upper Mississippi River Restoration Program. The National Science Foundation's "Research Experience for Undergraduates" Program Award Number:1559663 funded authors Casey McKean, Alaina Stockdill, Amber Lee, Frederick Forrest Miller, Killian Davis for data analysis and manuscript preparation.

**Competing interests:** The authors have declared that no competing interests exist.

## Author summary

The Upper Mississippi River System in the USA is a large, complex ecosystem that extends from about St. Paul, Minnesota, to the confluence with the Ohio River near Cairo, Illinois, and up the Illinois River. The system benefits from a water quality monitoring program for which consistent data have been collected since about 1986. We used a set of mathematical tools called 'topological data analyses' to help identify common water quality conditions or 'states' of this important river system. We also explored when and why the states have changed over the past 27 years. The tools identified five common water quality states, distinguished by nutrients and clear waters or muddy and turbid waters. State changes were commonly detected as seasonal transitions in the higher latitude reaches. The state changes also showed that water quality had improved or declined gradually over many years, depending on the river reach. River reaches named Pool 4 and Pool 8 had become clearer over time. Pool 13 became muddier over time with several undesirable state changes to a turbid-state. Three other river reaches were consistently muddy. Our study is a good example of how these topological data analysis tools may help others managing ecosystems by improving communications about what states exist and which are desirable. Coupling these tools with long-term data can also alert ecosystem managers to the ongoing, complex changes and vulnerability to help prioritize ecosystem rehabilitation.

## Introduction

Many ecosystems worldwide are undergoing severe and rapid changes and managing ecosystem states, and states transitions are a time-sensitive and difficult challenge. Large floodplain rivers are especially degraded and continue to experience major changes to hydrology and water quality [1,2]. Monitoring the transitions of ecosystem states with long-term data, and advanced analytical tools can provide useful information for protecting and rehabilitating ecosystems [3].

Ecosystem states are a recurring collection of biological and chemical attributes that arise from structuring processes and have some resilience to changes within its state space [4–6]. Characterizing and defining the state is always multidimensional, but typically only a few state variables are identified that properly characterize the ecosystem state. State variables are features of the biological community and physical environment that characterize the structure and functions of an ecosystem [6]. State transitions are either temporary or permanent changes to state variables that alter the ecosystem state. In contrast, regime shifts are substantial, persistent through time, and difficult to reverse with management action [7]. Identifying ecosystem states and the associated state variables is essential for describing the ecosystem and setting clear and reasonable ecological rehabilitation targets.

Ecosystem states are well defined in many types of ecological systems, but not for large rivers. Many ecosystems have at least two alternative states often defined by the lower trophic levels, such as the plant-dominated or phytoplankton-dominated states in lakes [8,9], marine kelp forest or kelp-deforestation [10], mesic grasslands or desertification [11], mesic grasslands or shrublands [12], and diverse coral reefs or macroalgal-dominated reefs [13]. A few examples of state transitions include the rehabilitation of aquatic vegetation and water quality in shallow lakes [9], altered disturbance regimes and subsequent rehabilitation in tallgrass prairies [14,15], and successive state shifts and a novel state arising from multiple stressors in a deep lake [16].

Ecosystem states have been conceptualized [17,18] but not well tested in rivers until recently [19–22]. The lack of research may be attributed to the difficulty in defining riverine ecosystem states because the multivariate methodology to tackle a state definition is complex and data are often sparse. The global scientific and management community calls for applying an ecosystem states framework to rivers, while acknowledging most rivers do not have sufficient data to properly evaluate states [7,18,23]. When sufficient data are available, identifying the states and drivers of the transition between states is key for ecological rehabilitation planning, post-project monitoring, and adaptive management [6].

Ecosystem state concepts are complex and multidimensional that ideally are informed by substantial data, advanced and novel analysis techniques for disentangling patterns, and expertise for their ecological interpretations. More analytical tools for big data that can reasonably predict ecosystem states and impending state transitions would help conserve, restore, and manage ecosystem resilience [3]. Topological data analysis (TDA) provides a powerful suite of tools for disentangling patterns and relationships among multidimensional data and presenting data visually [24]. Tools of TDA effectively have shown changes in the human gut following bacterial disturbance [25], predicted wildfire severity [26], and assessed vulnerability of residential homes to climatic changes [27]. Topological data analysis may be an ideal tool for many ecological applications because of its multidimensional framework [27], despite rare use in ecology today.

## Objectives and Hypotheses

We provided an analysis of ecosystem states and transitions using TDA tools to better characterize the Upper Mississippi River System (UMRS), which is an economically and environmentally significant ecosystem [2]. We hypothesized that this river would be in at least two states [18,23]: a "status-quo state" that is common and characterized by a broad range of turbidity and nutrient concentrations, and a "turbid-state" characterized by extremely high turbidity due to high suspended solids and chlorophyll *a* (Hypothesis 1). We expected to identify at least three state variables similar to regional shallow lake ecosystems [9] that include total phosphorus, suspended solids, and chlorophyll *a* (Hypothesis 2). We hypothesized that different river reaches were in different states (Hypothesis 3) given their differing water quality conditions and rehabilitation priorities [2,28]. We hypothesized TDA would detect an abrupt state transition around years 2007–2010 when aquatic vegetation rebounded and stabilized at high prevalence in several upper reaches and water quality conditions improved [19,29] (Hypothesis 4a). We further expected that TDA could also detect gradual water quality improvements based on the proportion of sites leaving the "turbid-state" and entering a "status-quo state" (Hypothesis 4b).

## Results

### How many ecosystem states were detected in the UMRS? Hypothesis 1

The number of ecosystem states varied depending on the algorithm chosen: The TDA Mapper showed two states, whereas the refined topological states algorithm showed five states. The TDA Mapper inputted 8 water quality variables collected from 69,307 sites over 27 years onto a single TDA structure (Fig 1). The TDA Mapper graph output had six connected components that included more than 99% of the data. Only <1% of the data points were excluded from the main TDA structure as outliers or noise (Fig 1 dark violet nodes). The main TDA structure had a shape of a "body" (containing ~90% of the data) and a "tail" (containing about 10% of the data), indicating two main ecosystem states (in support of Hypothesis 1).

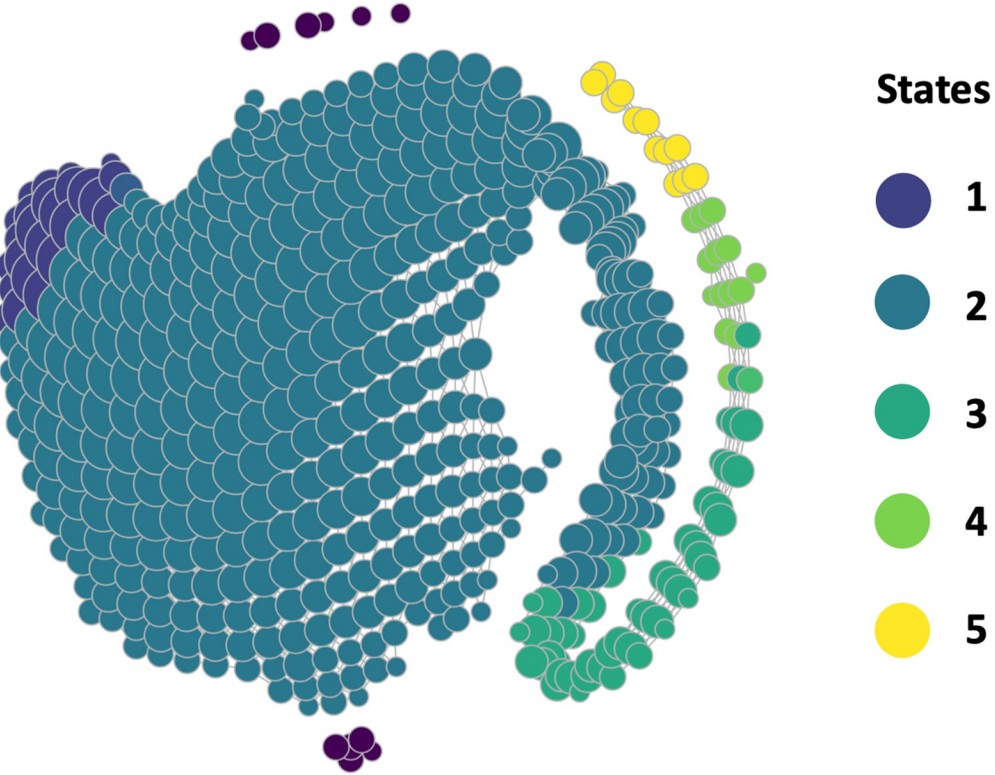

**Fig 1. Graphical output from topological data analysis (TDA Mapper) coupled with a density-based state assignment of the topological and ecosystem states for the Upper Mississippi River System, USA.** Data inputs included 8 water quality variables collected four times each year from 1993–2020 at a total of 69,307 sampling sites. The graph is composed of nodes (circles), which cluster sampling sites along the river based on water quality similarities, and the edges (lines) show statistical connections among nodes. An undirected graph (visualized by ignoring color scheme and focusing only on data shape) showed two topological states, indicated by the tail being less connected to the main body and thus having differing topology. The TDA Mapper was then transformed into a directed TDA Mapper graph using the topological states algorithm that further refined the shape into five states (color coded). The topological states may be conceptualized as ecosystem states, especially when further exploring data within the nodes (circles) to understand possible ecosystem state variables that classified each site within a node and distinguish among topological states. The dark purple nodes not connected to the main structure (<1% of sites) were outliers and not classified within a state.

The topological states algorithm further refined the two ecosystem states from the TDA Mapper (body and tail) outputs into five ecosystem states in a directed graph (Fig 1). States 1 and 2 were nested within the TDA Mapper body and States 3–5 were nested within the TDA Mapper tail. There were five, small transition nodes (meaning the sites were classified into both states simultaneously), which occurred between States 1 and 2 but could not be seen graphically in two dimensions because the nodes were small. Transition nodes were visible in a 3-dimensional TDA Mapper output, but we did not explore the transition nodes further because of the rarity of these transition nodes. States 1, 2, 3, 4, 5 contained 20%, 85%, 2%, <1%, and <1% of data across the six reaches and 27-year record, respectively.

### What were the state variables that defined and distinguished riverine ecosystem states? Hypothesis 2

State 1 may be considered a seasonal clear-water state because this state had the lowest total phosphorus, highest dissolved oxygen, lowest chlorophyll *a*, coldest water temperatures,

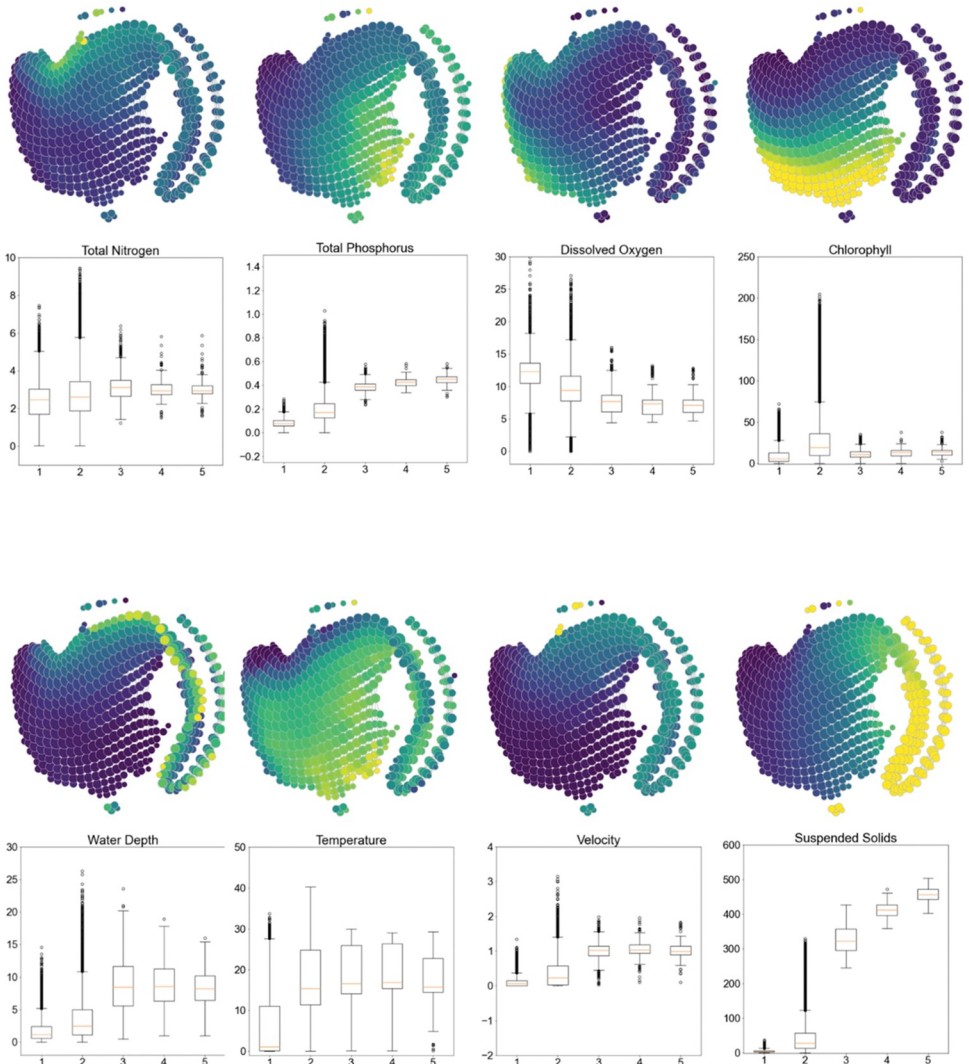

**Fig 2. Graphical outputs from topological data analysis (TDA Mapper) showing ecological gradients of eight water quality variables in the Upper Mississippi River System, USA ($n = 69,307$ sampling sites from 1993–2020).** The color coding represents relative values for each variable; dark violet colors are relatively low values, whereas greens and yellows are relatively high values. The boxplots below each TDA Mapper graph showed the data distributions for each ecosystem state (that corresponds to the states in Fig 1). Boxplots are standardized by boxes (containing the 25th, 50th (median), and 75th percentiles), the whiskers (containing the 25th and 75th percentiles), and the dots (statistical outliers). The x-axis on the boxplots represents the five topological and possibly ecosystem states (States 1–5). The y-axis represents variable units (total nitrogen = mg L$^{-1}$; total phosphorus = mg L$^{-1}$; dissolved oxygen = mg L$^{-1}$; chlorophyll $a$ = µg L$^{-1}$; water depth = cm; temperature = ˚C; velocity = m sec$^{-1}$; suspended solids = mg L$^{-1}$).

slowest water velocity, and lowest suspended solids relative to the other four states (Fig 2). State 1 was differentiated from State 2 principally by lower suspended solids (Fig 2). State 2 contained most sites (Tables 1 and 2; the so-called status-quo state), and the state variables differentiating State 2 from others included total phosphorus, chlorophyll $a$, and suspended solids (in support of Hypothesis 2; Fig 2). State 2 had the greatest variability, and the state variables had many statistical outliers. State 2 was further typified by lower total phosphorus but higher chlorophyll $a$ relative to States 3, 4, and 5.

States 3, 4, 5 within the structure's tail resembled the turbid-state. These three states were similar to each other for seven water quality variables (Fig 2) and were differentiated from one

**Table 1. The distribution of sites in the Upper Mississippi River System (column 1: River Reach name, field station city and state) that were classified into a topological/ecosystem state within years 1993–2019.** Table 1 shows the percentage (%) of sites in each state according to reach for comparisons of ecosystem states *among the six* reaches (i.e., columns sum to 100%).

| River Reach | State 1 | State 2 | State 3 | State 4 | State 5 |
|---|---|---|---|---|---|
| Pool 4 (Lake City, Minnesota) | 31 | 16 | 0 | 0 | 0 |
| Pool 8 (Onalaska, Wisconsin) | 35 | 17 | <1 | 0 | 0 |
| Pool 13 (Bellevue, Iowa) | 26 | 18 | <1 | 0 | 0 |
| Pool 26 (Brighton, Illinois) | 5 | 16 | 8 | 5 | 6 |
| Open River (Jackson, Missouri) | <1 | 17 | 90 | 95 | 93 |
| La Grange (Havana, Illinois) | 2 | 16 | <1 | 0 | <1 |

another by the relative concentrations of suspended solids (Fig 2). Compared to States 1 and 2, States 3–5 had greater total phosphorus, lower dissolved oxygen, greater water depths and velocities, and orders of magnitude greater suspended solids. States 3, 4, and 5 also had highest water depths and velocity. We removed total nitrogen from consideration as a state variable because it was the only variable that did not vary in data distribution among the five state classifications (Fig 2).

## Did reaches exist in different ecosystem states? Hypothesis 3

The six study reaches representing 2200 km of river were classified into different topological and ecosystem states (in support of Hypothesis 3). Different states were highlighted in different locations on the main TDA structure (Fig 3) and noted by the percentage of sites in each reach's state classification (Tables 1 and 2). Pool 4 (Minnesota), Pool 8 (Wisconsin), Pool 13 (Iowa), and La Grange Pool (Havana, Illinois; Illinois River) were within States 1 and 2 across the 27-year record. Therefore, Pools 4, 8, and 13 were principally categorized as within a status-quo state (State 2) characterized as having a broad range of ecological conditions but often included the clear-water state (27–39% sites within State 1 for these reaches). Pool 26 (Brighton, Illinois) existed in all five states, but State 2 was dominant (Tables 1 and 2 and Fig 3). Open River (Jackson, Missouri) was the principal reach within States 3–5. Open River was rarely in State 1, mostly within state 2, and 13% of sites were within the turbid states of States 3–5.

## Did reaches undergo state transitions within the past 27 years? Hypothesis 4

The six reaches varied substantially in their ecosystem state stability and the frequency of state transitions (Fig 4). Pool 26 and La Grange were stable within State 2 and occasionally State 1 and did not undergo state transitions. Pools 4, 8 and 13 displayed cyclic dynamics between State 1 and State 2 that were recurring annually due to seasonal changes in water quality. State

**Table 2. The distribution of sites in the Upper Mississippi River System (column 1: River Reach name, field station city and state) that were classified into a topological/ecosystem state within years 1993–2019.** Table 2 shows the percentage (%) of sites classified within a reach to compare the distribution of states *within each* reach. Sites sometimes can exist in more than one node, and therefore the percentage of sites within a state may sum to greater than 100% (i.e., rows sum greater than or equal to 100%).

| River Reach | State 1 | State 2 | State 3 | State 4 | State 5 |
|---|---|---|---|---|---|
| Pool 4 (Lake City, Minnesota) | 37 | 81 | 0 | 0 | 0 |
| Pool 8 (Onalaska, Wisconsin) | 39 | 78 | <1 | 0 | 0 |
| Pool 13 (Bellevue, Iowa) | 27 | 82 | <1 | 0 | 0 |
| Pool 26 (Brighton, Illinois) | 7 | 95 | <1 | <1 | <1 |
| Open River (Jackson, Missouri) | <1 | 87 | 9 | 3 | 1 |
| La Grange (Havana, Illinois) | 5 | 92 | <1 | 0 | <1 |

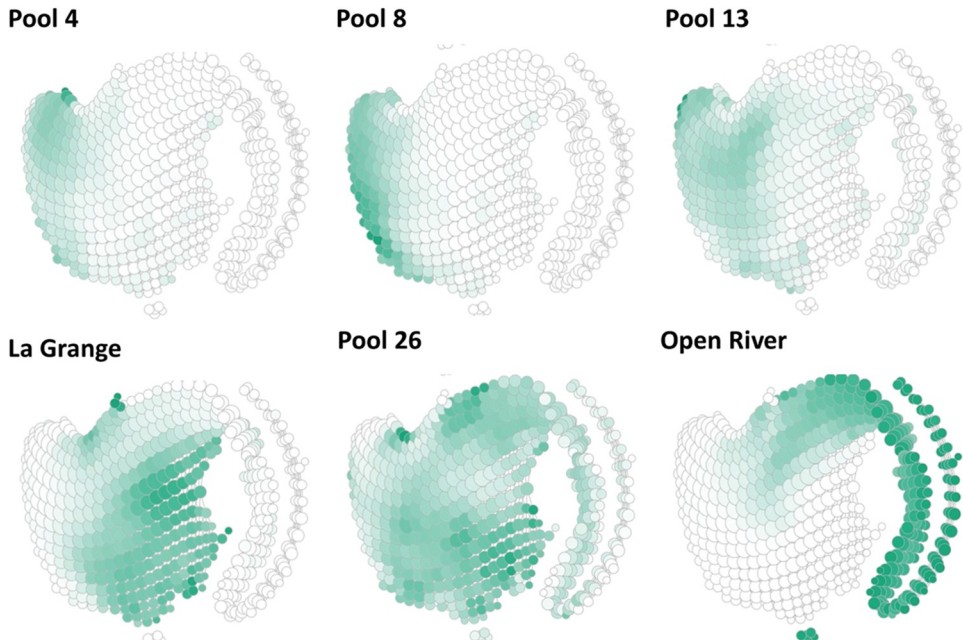

**Fig 3. Topological data analysis (TDA) showed spatial differences of water quality (eight variables; total nitrogen, total phosphorus, dissolved oxygen, chlorophyll *a*, water depth, water temperature, velocity, and suspended solids) among six reaches on the Upper Mississippi River System (years 1993–2020).** The green colors highlighted where sampling sites within each reach grouped onto the primary TDA Mapper structure, and the different highlighted locations indicated spatial differences among the six reaches. Dark green nodes had more sites within each node compared to the lighter green nodes. Reaches can be classified into specific ecosystem states by referencing their positions within Fig 1; for example, Pool 4 was within States 1 and 2 because the green highlights are on the left side of the TDA structure. As another example, Open River was principally within States 2–5 because the green highlights are on the right side of the TDA structure and its tail. Reaches can also be characterized by water quality state variables in Fig 2; for example, Open River is typically characterized by high velocity and suspended solids.

1 dominated in winter of most years and State 2 dominated in the other three seasons (S1 Fig). State 1 was uncommon but possible outside of winter, and State 1 was rare in summer (indicated by low proportion values of State 1 during summers). State 1 was seasonally dynamic as indicated by cyclic temporal proportion values that were greatest in winter, decreased by spring and summer, and then increased by the following autumn and winter (Fig 4).

Pool 4 and Pool 8 gradually shifted towards State 1 through time. The change detection function did not detect an abrupt state transition around years 2008–2010 in Pools 4 and 8 (refuting Hypothesis 4a). However, Pool 4 and Pool 8 had increasing minimum proportion values through time for State 1, especially after 2010 (Fig 4). The increasing minimum proportion values indicated State 1 (a clear-water state in winter) became more prevalent over time and thus water clarity during winters improved in those two reaches since 1993 and especially since 2010 (in support of Hypothesis 4b).

Pool 13 showed different patterns in state transitions (Fig 4). The increase in the proportion of minimum proportion values over time in State 1 were less pronounced than in Pools 4 and 8, indicating less increase in water clarity from decreases in suspended solids. Since 2015 in Pool 13, the previous pattern of transitioning to State 1 in winter had become less common and predictable compared to previous years. State 3 (a turbid state) was first entered in spring 2017 and again in spring 2019 at some Pool 13 sites, indicating decreased water clarity from increased suspended solids.

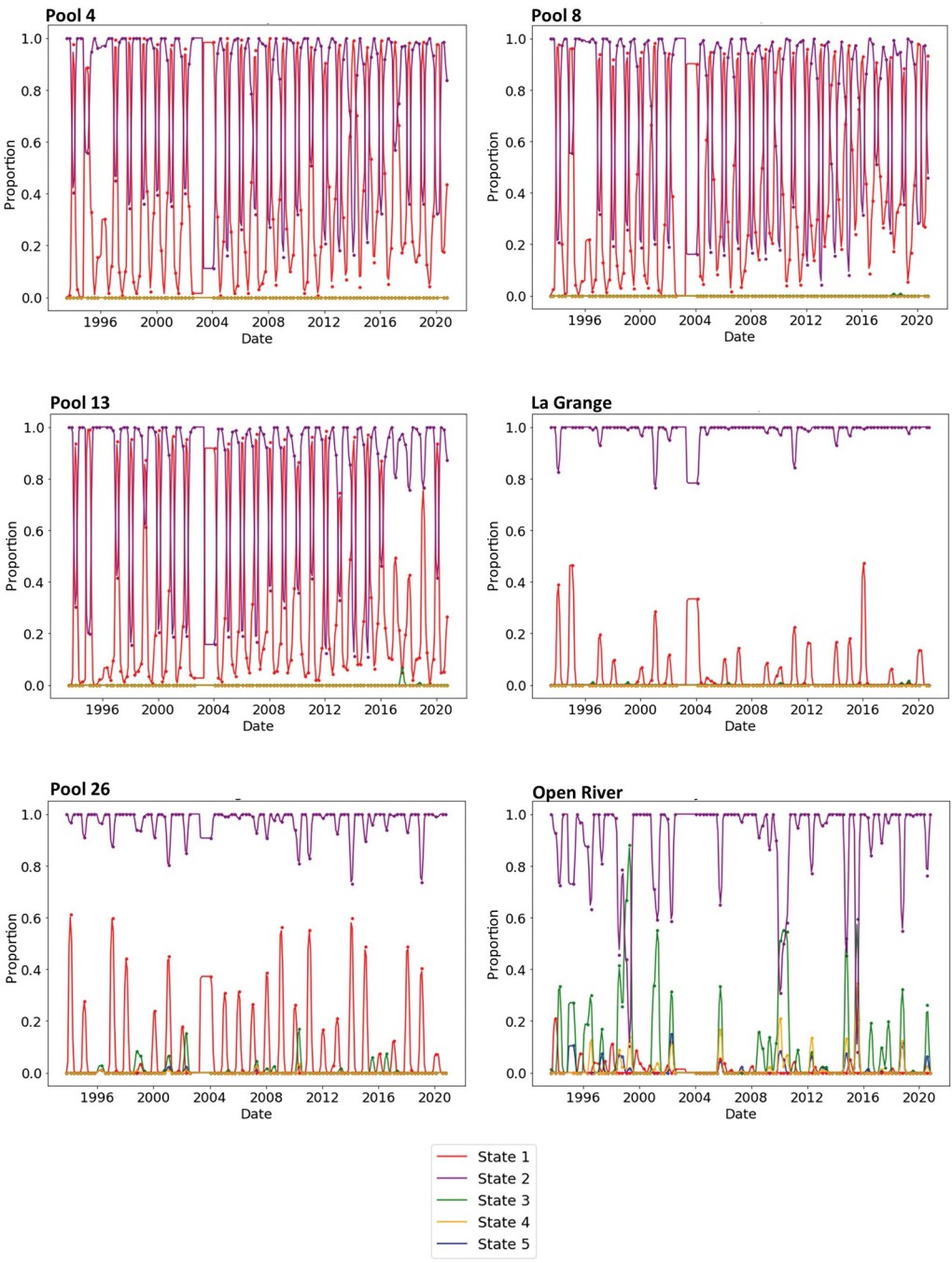

**Fig 4. Change detection plots showed the transitions in ecosystem states within six reaches of the Upper Mississippi River System from 1993 to 2020.** The proportion value (y axes) is calculated as the proportion of sites classified within each state each season. Occasionally, sites were classified into multiple states, and therefore the proportion sum may exceed 1.0. Points that fall directly on the year were collected in winter, followed by spring, summer, and autumn samples. Data from 2003 were missing due to logistical constraints for collection.

Open River (Jackson, Missouri) transitioned between State 2 and State 3 sporadically four times over 27 years. Transitions to State 3 generally only lasted one or two seasons before returning to State 2. The four state transitions in the Open River reach occurred across various

seasons and years (specifically, spring-summer 1999; winter-spring 2010; winter 2014; autumn 2015).

## Discussion

This is the first study to use topology tools of this type to quantify ecosystem states and state transitions in a large ecosystem. Our TDA detected patterns for multiple riverine states, the defining state variables, and differences among reaches in the UMRS (in support of our Hypotheses 1–3 and visualized in Figs 1–4). Our results expand the existing UMRS literature by using a multivariate, TDA approach to classify nearly 70,000 sites into multiple ecosystem states, identify water quality state variables, and explore state transitions. The TDA change detection analysis examined multivariate state transitions (Fig 4 and temporal proportion values), which scientists and managers to assess multiple correlated variables and how they vary synchronously in space and time. The TDA detected short-term state changes based on seasonality and episodic events and provided evidence of gradual, long-term changes due to water quality improvements over the recent three decades (Fig 4). We discuss how TDA may be used as a new technique for describing ecosystem states at multiple spatiotemporal scales, how the results can inform ecological rehabilitation and management, and how the state concepts tested with TDA methods may be transferred to other ecosystems that may exist in one or more ecological states.

### TDA as a novel technique for ecosystem states and transitions

Our application of TDA to the assess ecosystem state concepts was inspired by its prior use to explore state transitions in the human-gut microbiome [25,30]. Our study confirmed that TDA can be an effective method to identify ecosystem states and state variables from big data of large ecosystems. Our TDA results align well with previous studies of the UMRS that described large differences among reaches and water quality changes through time [2,19]. The TDA results supported another authors' previous hypothesis that the UMRS exists in two alternative states: status-quo (States 1 and 2) and turbid states (States 3–5) as defined by state variables of suspended solids and chlorophyll *a* [18]. State 2 had very high variability across all state variables that defines the status-quo state, meaning the water quality is highly variable across large spatiotemporal scales. Likewise, the main differentiation between the rare States 3–5 was based solely on concentrations of suspended solids, which may be better represented as an ecological gradient than firm state classification via TDA. Although we could have altered the TDA density algorithm to explore more sub-states within these five states, we did not have justification for refinement and wanted to provide a reasonable number of states that were clearly differentiated for aiding applied conservation [28].

The ecosystem states concepts, the TDA methods chosen to reflect ecosystem states, and the TDA results have some subjectivity in determining usefulness towards ecological understanding and management. For example, we suggested the TDA Mapper outputs of two alternative states (Fig 1) and the TDA algorithm refining two states into five states (Fig 2) are both valuable to display and guide resource management across the UMRS. Users can decide at which timescale a state is "stable." The change detection function we used statistically determined when states had transitioned, but the user would ideally determine that those results signify expected seasonal transitions (Figs 4 and S1), abrupt regime shifts (not detected in this study), or unmeaningful noise.

The TDA change detection function explored whether state transitions were abrupt and long-standing (i.e., an abrupt regime shift) or whether transitions were temporary state changes (i.e., transient dynamics) (Hypothesis 4). The TDA detected frequent transient

dynamics based on seasonality and episodic state changes (Fig 4) but did not detect abrupt state changes with sharp break points in time. Future TDA analyses could include data on submersed aquatic plant biomass or species composition [31] because aquatic plants are likely another state variable [18] that may reveal abrupt shifts around 2010 in support of our Hypothesis 4. Other TDA tools like "persistence homology" can detect specific change points [32], and thus may detect abrupt state transitions that are hypothesized to occur in rivers with long retention times [33] or following ecological rehabilitation [18]. Alternatively, the UMRS' ecology may be highly multidimensional and complex so that there was not a single point in time where major and abrupt changes happened, but rather changes were gradual over decades and most accurately portrayed with the change detection function used herein (Fig 4). The lack of abrupt regime shifts in the UMRS differs from those commonly observed within nearby shallow lake systems, which can transition to the alternative state within a few days and persist for years [8,9].

The TDA allows extraction of information at finer spatiotemporal scales; this was beyond the scope of this paper but could be explored in future work. For example, information from the data within the nodes can further describe how specific sites, seasons (S1 Fig), riverine habitats, and years are classified within each state to further ecological understanding and prioritize ecological rehabilitation. Sites classified within multiple states, transition nodes, or the TDA outliers could be further explored for unusual site characteristics and assessing site-level variability that affects ecosystem state.

## Ecosystem states results can guide ecological management & rehabilitation

Coupling ecosystem state theories with scientific models can provide scientific guidance for ecological rehabilitation and management [34,35]. Previously applied uses of state theory included predicting pending state transitions [36–38], managing systems for resilience and preventing undesirable transitions [37], and managing state variables to restore or protect resilience of desirable ecosystem states [9,39]. However, state classification and state transition models have been rarely used within large monitoring programs because of modeling complexities and the inability of institutions to practically apply model outputs [3].

We used big data (nearly 70,000 sites, 27 years, and eight variables) from long-term river monitoring [40], and these TDA results can guide structured decision making and vulnerability assessments in the UMRS. This is the first study in this large river to quantitatively identify ecosystem states, which is foundational for describing the system's ecological structures and functions. Having defined ecosystem states allows practitioners to communicate which states are desirable for preservation and which are undesirable and may warrant mitigation, as initiated in [18,28,41]. We identified the state variables (specifically total phosphorus, suspended solids, and chlorophyll *a*) that can guide decision making and actions for regulatory and management agencies by providing quantitative targets. Currently, specific regional water quality criteria are recommended for total phosphorus and chlorophyll *a*, but not suspended solids [42]. Specified targets for suspended solids concentrations are lacking, although water clarity remains a common restoration goal [28], and our TDA results echo the importance of suspended solids as a state variable.

The TDA's change detection can be a novel tool for assessing state transitions and vulnerability as new, long-term data are gathered. This function effectively signaled state changes in the UMRS due to seasonality, episodic events, and gradual changes to water quality (Fig 4). The state transition modeling in this system can continue to reveal gradual improvements to water quality and signal vulnerability to undesirable state transitions if States 3–5 become more prominent in certain reaches, such as Pool 13 (Fig 4). The TDA tool detected changes to

the variability of the number of sites within a state as the proportion values, and a change in time series variability can be an early warning indicator of ecological transitions [43]. The TDA could detect if more sites within Pool 13 continue to enter State 3 more frequently or persistently soon. The state variables of total phosphorus, chlorophyll *a*, and suspended solids identified for the UMRS are the same as those of lake ecosystems [8]; assessing vulnerability to state transitions using these variables could provide a warning for aquatic systems beyond the Mississippi and Illinois Rivers. Lastly, the states could become a categorical response variable in future models that aim to assess environmental drivers and threshold responses, although researchers would ideally carefully consider whether such information is better addressed with gradient analyses or state classification [44].

## Transferability of TDA to other ecosystems

Ecologists have emphasized the need to apply ecosystem states concepts to rivers worldwide, while acknowledging most rivers do not have sufficient data to properly evaluate states [2,7,17,18,23]. The UMRS has long-term data, and our results supported the concepts of alternative states in this particular river [18]. We suspect that similar ecosystem states based on water quality state variables can be found in other rivers, although we are unaware of other case studies to-date other than clear-water and turbid states based on aquatic plant abundance and water quality state variables [23]. The Regime Shifts Database [7] could serve as a data source to test our TDA ecosystem states framework in other rivers with similar water quality data (e.g., known hypoxia regime shifts within the Amite River, Brazo River, Chester River, and Indian River in the USA). For ecosystems without the big data available for the UMRS,

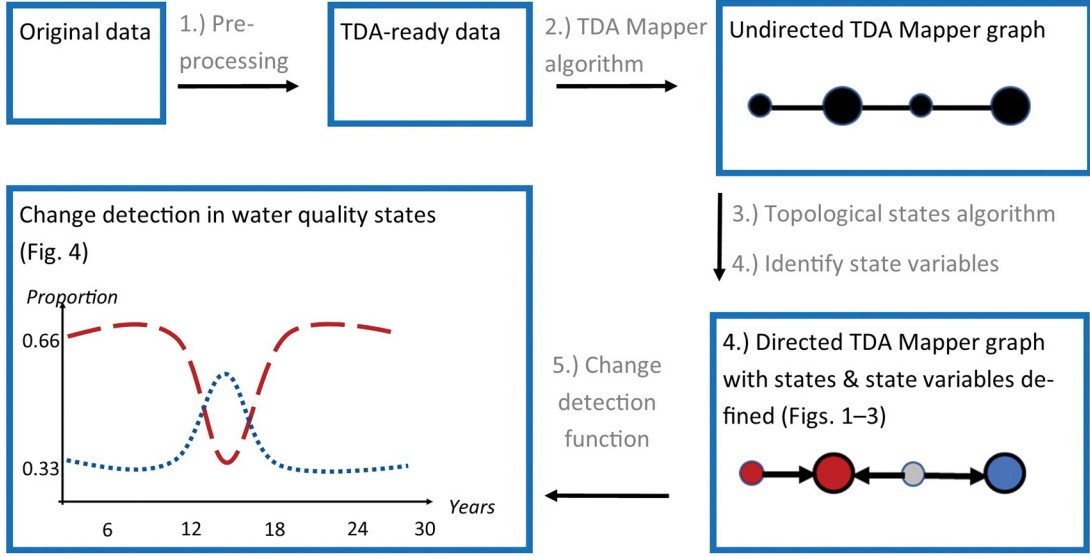

**Fig 5. Flow chart that summarized our topological data analysis (TDA) methods in sequential order (arrows and gray text) and the outputted products (boxes with black text and images) for quantifying ecological states and state transitions of the Upper Mississippi River System.** The original- and TDA-ready datasets and all analysis codes are available here (DOI: 10.5066/P976O6PP). In the "Undirected TDA Mapper" box, nodes and edges are connected to create a single TDA Mapper structure. In the "Directed TDA Mapper" box, the *example output* describes two local maxima nodes defining two states: a red state containing two nodes and a blue state containing one node. The gray node is a transition node, and the sites within the transition nodes are classified into both the red and blue state. In the "change detection" box, we provide a *hypothetical output* of where the dominance of two states (red and blue dashed lines, which are the proportion of sites within those states) temporarily transition from the dominant red state to the blue state around years 13 to 17.

monitoring system status and trends (like state transition warnings) could focus on the three state variables in aquatic systems: total phosphorus, chlorophyll *a*, and suspended solids. Transferability of the TDA approach (outlined in Fig 5) would be limited to ecosystems with sufficient data, although the number of sites and variables necessary for useful ecological assessments with TDA are not yet known.

Overall, TDA was a novel technique that informed many hypotheses regarding the UMRS as characterized by predictable ecosystem states, a few state variables, and state transitions. The TDA may provide an early indication of vulnerability to undesirable ecosystem transitions, such as Pool 13's increasingly frequent transitions to the undesirable state 3 with high suspended sediment loading. Testing ecosystem states in freshwater systems (beyond well-studied shallow lakes) is important, while being cognizant to the limitations and misuses of the theories [44]. The TDA is advantageous in that if spatiotemporal data are available, then the model outputs can be derived for a variety of spatial scales (e.g., site-, habitat-, reach-, and systemic-levels) and many temporal scales (seasonal- to multi-decadal) to match management objectives and a manager's geographical jurisdiction. Selecting state variables for TDA that are ecologically meaningful and having managers select those variables could aid in conservation efforts [45]. The novel coupling of ecosystem state concepts with multiple TDA tools revealed new information about the ecosystem states and state transitions of the UMRS, and this approach (Fig 5) may be transferred to any ecosystem with big data to help classify states and understand their vulnerability to state transitions.

## Methods

### Study Area, Ecosystem State Context, and Long-term Data

The UMRS flows ~2200 km from St. Paul, Minnesota, to the confluence with the Ohio River in Cairo, Illinois, and includes the Illinois River [2]. The locks and dams within the UMRS have created impounded areas upriver from each lock and dam to maintain a deep navigation channel for commercial vessels, but the locks and dams do not modify water residence time or stop flooding. Installation of the locks and dams in the late 1930s greatly modified the river's physical template. Aquatic plants were abundant after lock and dam construction and then were abruptly reduced in the late 1980s for reasons unknown [29]. Aquatic plants gradually returned and remained abundant since 2010 even under extended flood conditions from 2017–2019 [29]. We studied six study reaches named Pool 4, Pool 8, Pool 13, Pool 26, La Grange Pool, and Open River that span multiple environmental gradients like geomorphology, hydrology, and water quality [2], and had long-term data on water quality.

We assessed ecosystem states based on water quality data at 69,307 sampling sites. Data were collected four times annually (over a 2-week sampling episode) in each season from 1993 to 2020 across six study reaches. This is a 27-year consecutive record, except no sampling was done in 2003. The water quality data were collected using a standardized protocol whereby sites were selected by a stratified random sampling design with effort distributed proportionally across strata. Strata were defined based on enduring geomorphic features and included the main channel, contiguous backwaters, impounded areas upstream from the lock and dams, and lotic side channels [46].

Using ecological literature and *a priori* hypotheses (stated in the Introduction), we chose eight water quality variables thought to be important for defining ecosystem states and ensured the variables were not highly correlated (Pearson's $r < 0.45$). The eight variables were total nitrogen = mg L$^{-1}$, total phosphorus = mg L$^{-1}$, dissolved oxygen = mg L$^{-1}$, chlorophyll *a* = μg L$^{-1}$, water depth = cm, temperature = ˚C, velocity = m sec$^{-1}$, and suspended solids = mg L$^{-1}$. Although the TDA can incorporate more variables, we carefully selected eight variables based

on ecological relevance to state theory and the ability to manipulate the variable through habitat rehabilitation (e.g., water velocity and nutrient inputs may be manipulated; see [28]).

## Quantifying ecosystem states and the state variables

To quantify the ecosystem states, we used several TDA tools (Fig 5), which are dimension reduction techniques that interpret complex data (such as the multi-faceted water quality data in the UMRS) using the geometric and topological features of the data. All data have underlaying shapes that can be visualized and interpreted with TDA to find hidden properties (e.g., ecosystem states and state variables), recognize complex spatial patterns (e.g., how ecosystems may differ across space), and classify large data into meaningful clusters (e.g., ecosystem states based on topological states). We performed five main tasks (Fig 5): 1.) pre-processed the data, 2.) applied the undirected, TDA Mapper algorithm, 3.) performed a density-based state assignment and created a directed TDA Mapper graph, 4.) identified state variables, and 5.) created a change detection function and calculate temporal proportion values to detect state transitions over 27 years.

**1. Data Pre-processing.** The original water quality dataset (accessed 06/01/2021 from https://umesc.usgs.gov/data_library/water_quality/water1_query.shtml), had 204,345 rows (site data) and 133 columns (variables), and contained data from 1993 to 2020, except 2003. We filtered the data by stratified random sampling scheme (and filtered out "fixed sample sites," which composed ~60% of the dataset), surface water collections, and selected eight continuous variables (because TDA cannot effectively handle binary or ordinal variables). Negative values, which represent readings below the minimum detection limit and were <0.2% of the data, were set to the minimum non-negative value of that variable in the entire dataset, and quality flag (QF) codes of "0," "A," "8," or "64" were set to "NA." After we removed three unreasonable and extreme outliers (e.g., total nitrogen > 25 mg/L), the final data size was 69,307 rows by 8 columns including water quality variables and year. Three of the variables (total nitrogen, total phosphorus, and velocity), per the study design, had less data collected than the other variables. The TDA does not allow missing values; therefore, we compared multiple interpolation methods including kriging, inverse distance weighting of various distance thresholds, polynomial regression of various degrees, regression trees, and random forests. We chose a random forests algorithm as the top performing interpolation method (when compared to the mean absolute error to the other methods), which had > 93% accuracy for all interpolated variables. The other occasional missing values were imputed with the median value for each variable. We used robust scaling so each variable's differing measurement units were accounted for in the multivariate analysis, and robust scaling calculates the distance between input values using the median and interquartile range.

**2. The TDA Mapper Algorithm & Topological States Definition.** The Topological Data Analysis (TDA) Mapper algorithm visualizes high dimensional data by incorporating dimension reduction techniques and user-selected clustering algorithms to classify sites into topological states [47]. We suggest that topological states may indicate ecosystem states if the input variables are ecologically relevant and state variables are later revealed to classify sites within the TDA structure. The TDA Mapper algorithm takes in point cloud data and produces a "simplicial complex," which is an undirected graph that shows the data shape based on many nodes and edges (Fig 5). The nodes are clusters of sites that are similar based on the data (and the number of sites within each node can vary), and the edges signify interactions or connections among nodes. Nodes that are farther apart in 2-dimensional space indicate the sites are quite different. Essentially, the undirected TDA Mapper creates a single shape of all the data,

consisting of nodes and edges. We used Keppler Mapper (a TDA Mapper) written in Python (van Veen et al. 2019) on the TDA-ready dataset. The TDA Mapper algorithm was from [47].

The parameters of the TDA Mapper algorithm determine the shape of the data and are chosen by the user. The TDA Mapper parameters include filter function, size of the hypercubes, percent overlap, and clustering algorithm. As of 2022, there were not established methods for selecting the TDA parameter values of interest with the TDA Mapper, except see [48], so we explored multiple parameters and observed their effects on the simplicial complex. The TDA Mapper graphs using different parameters should be consistent and insensitive to relatively small changes in parameter values. To ensure the shape of the TDA graph persists for various parameter choices, we ran the TDA Mapper for many combinations of these parameter values: cube size ([219,100], [273.75, 125], [328.5, 150], [383.25, 175], [438, 200]) and percent overlap (30%, 40%, and 50%). The TDA Mapper graphs persisted through these parameter combinations and verified our final parameter selections were appropriate and that the TDA Mapper algorithm captured the true shape of the underlying data space.

The final parameters we chose included a principal component analysis (PCA) as the filter function, the cube size of [273, 125], and density-based spatial clustering of applications with noise (DBSCAN) as the clustering algorithm with the following justifications. The first two principal components (as determined from PCA) were used as the filter function. We used the following heuristics to determine the optimal cube size and percent overlap parameters like [25]: each node contains no more than 10% of the data, and at least 90% of the data is used in the main TDA Mapper structure that includes the body, tail, and outliers. In other words, at least 90% of the data is used to create the TDA Mapper and as much as 10% of data could be acceptably lost as noise. We quantified and reported in the results the amount of data expressed in the final TDA Mapper output. We used the ratio between the explained variances of the first two principal components of our filter function as the ratio of the components in the cube size parameter. We used the cube size of [273, 125]] because the PCA with those values explained the most variance. We also used 50% overlap, which was not a sensitive parameter when exploring 40–60% overlaps. For our clustering algorithm, we chose the default parameter 'DBSCAN' within Keppler Mapper. To determine DBSCAN's optimal minimum samples and epsilon parameters, we used the Elbow method that uses a minimum sample parameter of two times the number of data attributes. The epsilon values are selected by the following process:

1. Finding the average distance of each data point's k-nearest neighbors (kNN)

2. Sorting and plotting the average kNN distances for all data points

3. Finding the point of greatest curvature. Because eight water quality variables were used to produce the simplicial complex, we used 16 as our minimum samples. The epsilon value used was 1.5 and calculated using the Kneed package [49].

**3. Density-based state assignment.** Next, we use a topological states algorithm to refine the simplicial complex (i.e., the <u>undirected</u> TDA Mapper output) into a <u>directed</u> TDA Mapper graph, which further refines the data's shape into more topological states (Fig 5). We applied a density algorithm to the undirected TDA Mapper graph to produce a directed graph with defined states as adopted and modified from [25]. We used k-nearest neighbor (kNN) distance. Each node is assigned a density of

$$D(V) = \frac{n^2}{\sum_{i \in V} kNN(i, k)}$$

where $n^2$ counts for differing node sizes and $kNN(i,k)$ is the kNN distance of point $i$ defined as

$$kNN(i,k) = \frac{\sum_{j \in N(k)_i} d_{ij}}{k}$$

where $N(k)_i$ is the set of $k$ nearest points to the point $i$, $k$ is 10% of the number of samples in the dataset, and $d_{ij}$ is the distance between points $i$ and $j$. This density assignment allows converting the undirected TDA Mapper graph to a directed graph by replacing the edge between two nodes by an arrow pointing from the node with smaller density to the node with larger density. The "topological state definitions" in algorithm 2 and Fig 5 shows how a Mapper graph is converted to a directed graph to quantify local maxima nodes.

The directed graph is grouped into topological states (and conceptually considered ecosystem states herein), and these states are determined by *local maxima* nodes towards which every arrow connected them to points (Fig 5). In other words, the number of local maxima nodes is equal to the number of states detected. The distance between a node and a local maxima node is defined to be the smallest possible number of edges connecting them. A single water quality sampling site may sometimes be placed into more than one node, and thus classified in more than one state, if two nodes in different states have an edge between them and intersection of points. If a node is equidistant from two or more local maxima, that node is called a 'transition node' and is assigned to both local maxima (i.e., classified in both states). Sampling sites with more than one state classification or in transition nodes represented sites that are not easily classified by the topological state's definition.

**4. Identifying state variables.** A state variable is statistically considered a variable most positively associated with data clustering into a given topological state and the variable's data distribution is different from that of other states. We included eight water quality variables for the TDA to capture the multidimensionality of ecosystem state concepts. To determine state variables, we examined the gradients of each water quality variable displayed on the TDA Mapper structure in relation to where the classified state was defined on the TDA Mapper graph. We also compared the data distributions among the five state classifications using boxplots for each variable. If the median (50th percentile) and boxplot whiskers (25th–75th percentiles) did not overlap among the states, we considered that a likely state variable. We did not run formal statistical analysis for statistical significance among the five boxplots (e.g., analysis of variance among the states) because the large number of data points ($n = 69,307$) and degrees of freedom precluded that option due to risk of Type 1 error (i.e., false positive).

## Quantifying ecosystem state transitions

**5. Change detection function.** We tested whether states were stable or transitioned over 27 years. We analyzed the state composition of each reach during each season (autumn, winter, spring, and summer). On average, each season-reach combination contained 122 datapoints. We defined temporal proportion values for a reach-season-state pair as:

$$f(x_{state}) = \frac{x_{state}}{n}$$

where $x_{state}$ is the number of sites within a reach classified within a particular state during that season, and $n$ is the total number of sites in any state collected during that reach-season combination. A hypothetical example showing a single state transition between two states based on the temporal proportion values is in Fig 5.

## Supporting information

**S1 Fig. Graphical output from topological data analysis (TDA Mapper) showing a strong seasonal component to the water quality data for the Upper Mississippi River System, USA.** Data inputs included 8 water quality variables collected four times each year during 1993–2020 at a total of 69,307 sampling sites. The graph is composed of nodes (circles), which cluster sampling sites along the river based on water quality similarities, and the edges (lines) show statistical connections among nodes. We highlighted in color the seasons in which sites were sampled. Seasonality can be connected to ecosystem states by referencing Fig 1 positions; for example, winter is typically characterized by State 1.
(DOCX)

## Acknowledgments

We sincerely thank Gregory Sandland, James Pierce, Roger Haro, and Richard Erickson for REU Program leadership and support in 2021. We thank Douglas Krouth (University of Wisconsin–La Crosse) for initiating the first TDA Mappers. This research was performed in part using computational resources supported by the Academic & Research Computing group at Worcester Polytechnic Institute. Any use of trade, firm, or product names is for descriptive purposes only and does not imply endorsement by the U.S. Government.

## Author Contributions

**Conceptualization:** Danelle Marie Larson, Wako Bungula.

**Data curation:** Danelle Marie Larson, Wako Bungula, Casey McKean, Alaina Stockdill, Amber Lee, Frederick Forrest Miller, Killian Davis.

**Formal analysis:** Danelle Marie Larson, Wako Bungula.

**Funding acquisition:** Danelle Marie Larson, Wako Bungula.

**Investigation:** Danelle Marie Larson, Wako Bungula, Casey McKean, Alaina Stockdill, Amber Lee, Frederick Forrest Miller, Killian Davis.

**Supervision:** Danelle Marie Larson, Wako Bungula.

**Visualization:** Danelle Marie Larson, Wako Bungula, Casey McKean, Alaina Stockdill, Amber Lee, Frederick Forrest Miller, Killian Davis.

**Writing – original draft:** Danelle Marie Larson, Wako Bungula.

**Writing – review & editing:** Danelle Marie Larson, Wako Bungula, Casey McKean, Alaina Stockdill, Amber Lee, Frederick Forrest Miller, Killian Davis.

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
