## [Decision Letter · Decision Letter 0]

24 Jan 2023

Dear Dr. Larson,

Thank you very much for submitting your manuscript "Quantifying ecosystem states and state transitions of the Upper Mississippi River using topological data analysis" for consideration at PLOS Computational Biology.

As with all papers reviewed by the journal, your manuscript was reviewed by members of the editorial board and by several independent reviewers. In light of the reviews (below this email), we would like to invite the resubmission of a significantly-revised version that takes into account the reviewers' comments.

We cannot make any decision about publication until we have seen the revised manuscript and your response to the reviewers' comments. Your revised manuscript is also likely to be sent to reviewers for further evaluation.

Sincerely,

Eric Lofgren, MSPH, PhD

Academic Editor

PLOS Computational Biology

Natalia Komarova

Section Editor

PLOS Computational Biology

Reviewer's Responses to Questions

**Comments to the Authors:**

Reviewer #1: The article is well organized, and the method section is clearly described. Can you please clarify the following questions:

1) In line 376: "To address missingness, we compared seven interpolation methods and utilized a random forests algorithm as the top preforming method"-- Correct "preforming" ->"performing". Can you please show the top performance value and which metric you used as the performance metrics? Also what other methods are used along with random forest?

2) In line 401: "As of 2022, there were not established methods for selecting parameters". Persistence homology can be used to find the appropriate overlapping value and stable TDA object. The supplementary part of this paper (https://ieeexplore.ieee.org/document/8880514) shows how persistence homology helps to find a stable TDA object, and you can refer to this paper.

Reviewer #2: review is uploaded as an attachment

Reviewer #3: Review of Larson et al. Quantifying ecosystem states and state transitions of the Upper Mississippi River using topological data analysis.

The authors present the results of a novel application of topological data analysis (TDA) – its use to assess a large data set for evidence consistent with the existence of differing ecological states. The quantitative method was able to classify data from ~70K site visits as belonging to one of 5 classes. The authors interpret those results as indicating the classes represent 5 ecosystem states. Starting with eight candidate variables, the analysis indicated that the primary variables determining the ecosystem classification were: suspended solids, chlorophyll a and total phosphorus. Temporal patterns were also assessed for evidence of abrupt changes in the ecological state; results suggested that changes were better described as gradual than abrupt. The quantitative results are for the most part clearly explained (exceptions noted in my attached comments), are consistent with other studies of this system, and new insights. The figures and text provide reasonable support for the quantitative results. As described in my general comments below, there are some aspects of the conceptual interpretation of those quantitative results that need to be better explained and possibly more carefully considered. I also provided detailed minor comments on the ms pdf (uploaded as an attachment) that should be considered during revisions. Assuming that these comments can be reasonably addressed, the manuscript will provide an interesting example of a novel application of an existing quantitative method to address questions in aquatic systems and more generally in ecosystem ecology.

General Comments

1. The authors need to clearly define how they are using “ecosystem state”. In much of the paper is seems to be used to mean a collection of certain conditions in the ecosystem. I think that is reasonable. In other places it is used interchangeably with “alternate regimes”, and in a few places the term “alternate regimes” is used. I don’t think it is correct to refer to regular, annual seasonal changes as “alternate regimes” as the term is usually used. Providing a clear statement of how “ecosystem state” is being used in this ms and a citation or two in support of that use would clear this up.

2. Related to comment #1, the likely scenario that all (or nearly all) of the state 1 observations occurred in winter is treated irregularly throughout the ms. In places it is explained clearly; in other places the text leaves open the possibility that these observations could represent a regime shift similar to those seen during the growing season in other shallow aquatic systems. For example, on line 252, the text suggests that the clear vs turbid states detected by TDA are the typical clear/turbid states observed in shallow aquatic ecosystems.

3. Visually it seems like a substantial portion of state 2 could be considered part of the tail rather than the body. How is the tail/body border determined? This is especially relevant regarding interpretation of data from the Open River study area. A related question: Does requiring the parameter specifications to produce a result where at least 90% of the data 90% of the data must be in the main body have the potential to affect the # of ecosystem states that are detected? If so, should this be considered in the interpretation of your results?

4. A brief description of the stratified random sampling design, the decision to collapse the data set across all years, and the implications of both would aid the reader in interpreting the results. This would also provide useful context for the text beginning line 339.

5. Topological figures have x, y and z(color) dimensions. Interpretation of color is explained. Is there meaning that can be inferred from x and y dimensions? For example, does the relative location within the “body” relative to the “tail” indicate anything? Also, there a many more samples than circles in the figures, how many samples does each circle represent?

**Have the authors made all data and (if applicable) computational code underlying the findings in their manuscript fully available?**

Reviewer #1: None

Reviewer #2: **No: **draft says: Our final datasets and analysis scripts will be archived in GitLab (DOI: XXXXXXXX) following peer review

Reviewer #3: Yes

PLOS authors have the option to publish the peer review history of their article (what does this mean?). If published, this will include your full peer review and any attached files.

Reviewer #1: No

Reviewer #2: No

Reviewer #3: **Yes: **Jeffrey N Houser
---

## [Decision Letter · Decision Letter 1]

12 Apr 2023

Dear Dr. Larson,

Thank you very much for submitting your manuscript "Quantifying ecosystem states and state transitions of the Upper Mississippi River System using topological data analysis" for consideration at PLOS Computational Biology. As with all papers reviewed by the journal, your manuscript was reviewed by members of the editorial board and by several independent reviewers. The reviewers appreciated the attention to an important topic. Based on the reviews, we are likely to accept this manuscript for publication, providing that you modify the manuscript according to the review recommendations. Please pay special attention to the remaining areas of ambiguity that have been identified, as well as a careful reading of the entire manuscript for clarity.

Sincerely,

Eric Lofgren, MSPH, PhD

Academic Editor

PLOS Computational Biology

Natalia Komarova

Section Editor

PLOS Computational Biology

Reviewer's Responses to Questions

**Comments to the Authors:**

Reviewer #1: This is an excellent applied manuscript on TDA and a new inspiration to the TDA community.

Reviewer #2: The additions and edits the authors made have improved the clarity of the methods and motivations, and the additional discussion has better placed this study in context with respect to its relevance for other rivers. I would still suggest the authors proofread carefully! But otherwise I think the study stands as a nice application to an important dataset of an analytic method that is useful for synthesizing large-scale data into meaningful and actionable conclusions.

Reviewer #3: The authors have substantially revised their previously submitted manuscript. Most of my previous comments have been satisfactorily addressed. My remaining comments are provided using the adobe comments tool in the attached pdf.

**Have the authors made all data and (if applicable) computational code underlying the findings in their manuscript fully available?**

Reviewer #1: Yes

Reviewer #2: Yes

Reviewer #3: Yes

PLOS authors have the option to publish the peer review history of their article (what does this mean?). If published, this will include your full peer review and any attached files.

Reviewer #1: No

Reviewer #2: No

Reviewer #3: **Yes: **Jeffrey N Houser

Figure Files:

Data Requirements:

Reproducibility:

References:

---

## [Editor Report · Decision Letter 2]

2 May 2023

Dear Dr. Larson,

We are pleased to inform you that your manuscript 'Quantifying ecosystem states and state transitions of the Upper Mississippi River System using topological data analysis' has been provisionally accepted for publication in PLOS Computational Biology.

Best regards,

Eric Lofgren, MSPH, PhD

Academic Editor

PLOS Computational Biology

Natalia Komarova

Section Editor

PLOS Computational Biology

---

## [Editor Report · Acceptance letter]

24 May 2023

PCOMPBIOL-D-22-01462R2 

Quantifying ecosystem states and state transitions of the Upper Mississippi River System using topological data analysis

Dear Dr Larson,

I am pleased to inform you that your manuscript has been formally accepted for publication in PLOS Computational Biology. Your manuscript is now with our production department and you will be notified of the publication date in due course.

With kind regards,

Timea Kemeri-Szekernyes
